# Marinated Anchovies (*Engraulis encrasicolus*) Prepared with Flavored Olive Oils (Chétoui cv.): Anisakicidal Effect, Microbiological, and Sensory Evaluation

**Najla Trabelsi** [1,*] , **Luca Nalbone** [2] , **Ambra Rita Di Rosa** [2] , **Abdelaziz Ed-Dra** [2,3] , **Salma Nait-Mohamed** [1] , **Ridha Mhamdi** [1] , **Alessandro Giuffrida** [2] and **Filippo Giarratana** [2]

1   Center of Biotechnology of Borj Cedria, LR15CBBC05 Olive Biotechnology, Hammam Lif 2050, Tunisia; salma.nait@cbbc.rnrt.tn (S.N.-M.); ridha.mhamdi@cbbc.rnrt.tn (R.M.)
2   Department of Veterinary Science, University of Messina, Polo Universitario dell'Annunziata, Viale Palatucci snc, 98168 Messina, Italy; lnalbone@unime.it (L.N.); ambra.dirosa@unime.it (A.R.D.R.); abdelaziz_iaa@yahoo.fr (A.E.-D.); agiuffrida@unime.it (A.G.); fgiarratana@unime.it (F.G.)
3   Team of Microbiology and Health, Laboratory of Chemistry-Biology Applied to the Environment, Moulay Ismail University Faculty of Sciences, Zitoune Meknes 50070, Morocco
*   Correspondence: najla.trabelsi@cbbc.rnrt.tn

**Abstract:** To meet the food demand of future generations, more sustainable food production is needed. Flavored olive oils (FOOs) have been proposed as natural additives to ensure food safety and quality through a more sustainable approach. The chemical composition and antioxidant potential of two different olive oils flavored, respectively, with cumin (Cm) and with a mixture of parsley, garlic, and lemon (Mix) were investigated. Cm-FOO and Mix-FOO were tested against *Anisakis* both in vitro and ex vivo through an exposure test of anchovy fillets experimentally parasitized with *Anisakis* larvae. Microbiological and sensory analysis were carried out on marinated anchovy fillets exposed to both FOOs to evaluate their effects on the shelf life and their sensory influence. The addition of herbs and spices did not affect the chemical composition of the olive oil (free acidity, UV absorbance, and fatty acid composition). Only Mix showed antioxidant activity, while Cm had no effect in this regard. Cm-FOO and Mix-FOO devitalized the *Anisakis* larvae both in vitro within 24 h and ex vivo after 8 and 10 days of exposure, respectively. The results of microbiological analyses showed that FOOs inhibited the growth of typical spoilage flora in the marinated anchovies without negatively affecting their sensory characteristics, as observed from the sensory analysis.

**Keywords:** cumin; parsley; garlic; lemon; *Anisakis*; olive oil; *Olea europaea* L.; marinated anchovy

## 1. Introduction

The current world population is estimated to rise to nearly nine billion by 2050, with a consequent increase in food demand [1]. To ensure enough food for future generations, more sustainable food production and changes in our eating habits are needed. The current way of producing, supplying, and consuming food has raised concerns for the outcomes of climate change, biodiversity loss, social inequalities, land degradation, poverty, hunger, and malnutrition [2,3]. Achieving a sustainable food production system and reducing food waste are major global challenges to produce equal, safe, healthy, and low environmental impact foodstuffs [4,5]. Therefore, food safety and sustainability are an essential paradigm for the preservation of the present and future public health [6].

Throughout history, herbs and spices have been added to foods to enrich their sensory characteristics and nutritional value and to extend their shelf life [7,8]. Currently, plant extracts, spices, and essential oils are used in the food industry as "green" alternatives to synthetic additives for extending shelf life and for their effects against foodborne pathogens [8].

In this regard, the ancient practice of adding herbs and spices to olive oils is still widespread and an object of research, considering their potential application as flavoring and preservative agents [9,10]. Over the years, several methods have been proposed to aromatize olive oil (infusion of natural material, addition of essential oils, cold pressing of olives with spices, and herbs or vegetables), which variably affects the acceptability, oxidative stability, quality parameters, and presence of bioactive compounds in the flavored oils [11,12].

The bioactive compounds (such as phenols, flavonoids, carotenoids, and anthocyanin.) normally present in the olive oil are responsible for antioxidant, antimicrobial, antiparasitic, antiviral, anti-inflammatory, antiulcerous, and anticarcinogenic properties, in addition to other several beneficial effects on human health [13,14].

Recently, the antiparasitic effect was reported for several flavored olive oils (FOOs) against *Anisakis* larvae [10,15]. The latter belong to a group of nematodes called anisakids responsible for the fish-borne zoonotic disease anisakidosis, caused by the consumption of raw or undercooked parasitized fish and cephalopods [16]. To date, the cases of human anisakidosis are increasing worldwide due to the growing consumption of raw (sushi, sashimi, etc.) and marinated fishery products [17,18]. For this reason, freezing treatment is used worldwide as a preventative treatment in order to kill viable parasites in fishery products intended to be consumed raw or almost raw [19]. Since freezing may affect the sensory characteristics of the products [20], alternative technological treatments have been proposed to obtain an equivalent effect [21,22]. Cinnamon-FOO (at 5%) and Cumin-FOO (at 5%) were tested against *Anisakis* larvae experimentally encysted in marinated anchovy fillets, showing a high efficacy [10]. However, results of the sensory analysis showed that Cinnamon-FOO negatively influenced the typical sensory properties of marinated anchovy, while Cumin-FOO resulted in a pleasant experience for all the panelists. Unsurprisingly, cumin is traditionally used in the fish dishes of North African and Middle Eastern countries, widely appreciated for its characteristic flavor and aroma typical of food from these countries.

The use of FOOs is conditioned by their influences on the organoleptic properties of food, not always appreciated by consumers. Therefore, sensory evaluation is crucial to evaluate their potential applicability. In recent years, much research has been performed to substitute the perception of human senses with "artificial sensors", instruments providing signals related to the sensory attributes and used to design a new sensory fingerprint model of food and offer a potential alternative for discrimination of multiple samples [23]. In this regard, advances in sensor technology, electronics, biochemistry, and artificial intelligence led to the development of instruments such as the electronic nose (e-nose), electronic tongue (e-tongue), and computer vision systems (CVSs) capable of measuring and characterizing the flavor, color, and chemical components of various products [24].

Against this background, in the present study, the anisakicidal effectiveness, sensory influences, and antibacterial activity of two different FOOs were evaluated in the marinating process of anchovy fillets. Their chemical composition and potential antioxidant activity were also investigated. The FOOs tested were obtained from different herbs and plants traditionally used in Mediterranean cuisine for the domestic preparation of fish dishes: (i) a mixture of parsley, fresh garlic, and lemon (Mix), widely used in European countries (Italy, Spain, Portugal, Greece, etc.) and (ii) Cumin (Cm), a spice broadly used in North African countries and the Middle East.

## 2. Materials and Methods

### 2.1. Experimental Plan

The present study was carried out in four main steps: (i) Preparation of the flavored oils and determination of their chemical composition: once the collected olives, herbs, and spices were mixed together through a physical treatment to obtain the Mix-FOO and Cm-FOO, the chemical composition of each flavored oil was determined by gas chromatography–mass spectrometry to assess the most representative compounds;

(ii) Anisakicidal activity: in vitro and ex vivo experiments were set up to assess the possibility of using Mix-FOO and Cm-FOO to manage the *Anisakis* risk in the industrial marinating process of the anchovy fillets. In the in vitro experiment, *Anisakis* larvae type I were directly treated with each FOO. In the ex vivo experiment, fresh anchovy fillets were experimentally parasitized by *Anisakis* larvae, then marinated at refrigeration temperature and treated with both FOOs at room temperature. The larval viability was monitored under a stereomicroscope (Leica M 205 C, Leica Camera AG, Wetzlar, Germany) at regular time intervals according to the criteria of Hirasa and Takemasa [25], assigning the following scores: 3 (viable), 2 (reduction of mobility), 1 (mobility only after stimulation), and 0 (death). Larvae were considered dead when no mobility for 5 min was observed under stereomicroscope in a saline solution (0.9% NaCl); (iii) Microbiological analysis: the activity of Mix-FOO and Cm-FOO as preservative agents to extend the shelf life of marinated anchovy fillets was investigated by assessing the antimicrobial effects on the growth of spoilage bacteria. Usually, marinated anchovy fillets in oil are marketed at refrigeration temperature; however, in this experiment, marinated anchovy fillets were kept at room temperature under the most favorable growth conditions for spoilage bacteria; (iv) Sensory evaluation of marinated anchovy fillets: the sensory influence of Mix-FOO and Cm-FOO on the organoleptic characteristics of the marinated anchovy fillets was evaluated through an instrumental sensory analysis, which included the e-tongue, e-nose, and e-eye.

*2.2. Preparation of the Flavored Oils and Determination of Their Chemical Composition*

2.2.1. Sample Collection and Process of Oil Aromatization

The virgin olive oil (VOO) and flavored olive oils were obtained from olive fruits (*Olea europaea* L.) cultivar Chétoui, picked manually from olive groves in the region of Takelsa (Nabeul, Tunisia). The lemon zest (*Citrus limon*) used for oil aromatization was obtained from plants located in the region of Takelsa. The seeds of cumin (*Cuminum cyminum*), parsley leaves (*Petroselinum crispum*), and fresh garlic (*Allium sativum*) were bought in a local market (Nabeul, Tunisia). Olive fruits were picked in 2018 and the oils were obtained within 24 h of harvesting using a laboratory scale mill. The VOO was obtained as follows: olive fruits were washed with potable water and crushed with a hammer mill (MM-100, Type mc2). The obtained paste was then mixed with a thermo mixer (TB-100, mc2) at room temperature for 30 min and finally centrifuged with a centrifugal machine (CF-100, mc2). The FOOs were obtained by adding the aforesaid spices at 100 g/kg (10%) to the homogeneous batches of olive fruits before pressing. The choice of spice, herb, and plant concentration was based on a consumer survey to establish the most appreciated concentration that did not negatively affect the oil taste. Once obtained, the VOO and FOOs were transferred into dark glass bottles and stored in the dark at 4 °C until analysis.

2.2.2. Quality Parameters

The assessed quality parameters of the VOO oil and FOO were: free acidity (FA), peroxide value (PV), and the specific UV absorbance conventionally indicated by $K_{232}$ and $K_{270}$. These parameters were carried out according to the European official methods of analysis [26].

2.2.3. Antioxidant Potential

Extraction and Determination of Biophenols

The total biophenols of the polar extracts of VOO and FOOs were determined following the method proposed by Dewanto et al. [27], with some modifications, and precisely by adding 0.125 mL of the phenolic oil extract to 0.125 mL of Folin–Ciocalteu reagent and 1.250 mL of a sodium carbonate aqueous solution (7% *w/w*). The final volume was brought up to 3 mL with distilled water. Then, the obtained mixture was stirred and kept in the dark for 90 min. Absorption at 765 nm was measured using a spectrophotometer (Carry 60 UV-Vis, Agilent Technologies, Bayan Lepas, Malaysia). Total phenolic content

was expressed as mg gallic acid equivalents (GAE)/kg of oil through the calibration curve with gallic acid. The calibration curve range was 0–1000 mg/mL.

In Vitro Antioxidant Activities: DPPH Test

VOO and FOOs were tested at different volumes (5, 10, 20, 40, 80, and 100 μL). These volumes were then adjusted to 1 mL by adding ethanol to Tween 20 (5 μL) and a 2,2-diphenyl-1-picrylhydrazyl radical methanolic solution (DPPH, 0.5 mM). The obtained mixtures were stored in the dark at room temperature for 1 h [28]. The negative control (CTL) was prepared as above without an oil sample. The absorbance was monitored at 515 nm, and radical scavenging was expressed as IC50 (μL/mL). The IC50 value denotes the volume (μL) of oil required to scavenge 50% of the DPPH free radical. The antioxidant potential is inversely proportional to IC50 value. The ability to scavenge the DPPH radical was calculated using the following equation:

$$\text{DPPH. scavenging effect (\%)} = [(A_0 - A_1)/A_0] * 100 \tag{1}$$

where $A_0$ and $A_1$ are the absorbance of the CTL and the sample at 60 min. All samples were analyzed in triplicate.

Fatty Acids Composition

In a 5 mL screw top test, 0.1 g of each oil sample was mixed with 3 mL of hexane. Then, 0.5 mL of a methanolic potassium hydroxide solution was added to the previous mixture and shaken vigorously for 30 s. The samples were left to stratify, and the upper layer containing the methyl esters of fatty acids (FAME), once clear, was decanted, ready for injection into the gas chromatograph.

The methyl esters prepared were analyzed using gas chromatography (7890B GC system, Agilent Technologies, Shanghai, China), equipped with a capillary column (Agilent CP6173) of 50 m × 250 μm × 0.2 μm film thickness fused with a Silica capillary column (Innowax). An injection volume and mode of 1 μL and split (100:1) were used. The GC oven temperature was programmed to increase from 50 °C to 240 °C at a rate of 10 °C/min and finally held for 6 min. Both the injector and detector were maintained at 230 °C. Helium was used as the carrier gas (column flow 1 mL/min). Fatty acids were identified by comparing their relation times with those of the standards compounds.

### 2.3. Anisakis and Anchovy Collection

The Anisakidae larvae were all collected from several specimens of silver scabbard fish (*Lepidopus caudatus*, Euphrasen, 1788) caught in the Straits of Messina (Sicily, Italy, FAO area 37) within 8 h of each experiment. The silver scabbardfish was selected since it is often parasitized by the nematode, due to being at the very top of the marine food chain and its eating behavior [29,30]. The celomatic cavity and inner organs of the fish were examined to detect Anisakidae larvae. The collected nematodes were all rinsed three times with a sterile saline solution (NaCl 9 g/L) and then carefully observed under the stereomicroscope (Leica M 205 C) to ensure their viability and belonging to the L3 *Anisakis* larvae type I, based on the morphological characteristics reported by Murata et al. [31]. Fresh store-bought anchovies (*Engraulis encrasicolus*; Linneo, 1758) were collected from local markets (Messina, Sicily, Italy) and processed within 18 h of harvesting. Each anchovy was gutted, headed, and reduced into fillets just before its use.

### 2.4. Anisakicidal Activity

#### 2.4.1. In Vitro Larvicidal Activity

A total of 180 *Anisakis* larvae were tested to evaluate the in vitro larvicidal activity of both Mix-FOO and Cm-FOO. For each test, 20 parasites were placed in a plastic petri dish (diameter 90 mm) in 20 mL of each FOO kept at controlled room temperature (20 ± 1 °C). *Anisakis* larvae placed in VOO were used as CTL and maintained under the same experimental conditions. All the experiments were carried out three times in separate conditions

and at different times, for a total of 60 *Anisakis* larvae tested for each oil. Larvae were macroscopically examined and observed under stereomicroscope (Leica M 205 C) to evaluate their viability at regular time intervals after 4, 8, 12, 16, 20, and 24 h. The normalized mean viability (mean value of the viability score, expressed as a percentage) was calculated according to the method proposed by Giarratana et al. [32] by using the following equation:

$$\text{Normalized viability score (\%)} = (\text{Mean viability score} * 100)/3 \qquad (2)$$

where mean viability score = the mean value of the *Anisakis* viability according to the criteria of Hirasa and Takemasa [25], assigning the following scores: 3 (viable), 2 (reduction of mobility), 1 (mobility only after stimulation), and 0 (death). The LT100 (lethal time required to kill 100% of parasites) and LT50 (time required to kill 50% of parasites) were calculated using IBM SPSS statistical software (version 24.0).

### 2.4.2. Ex Vivo Larvicidal Activity in Anchovy Marinating Process

After the in vitro test, the anisakicidal effectiveness of both Mix-FOO and Cm-FOO was tested in the industrial marinating process. The experiment was set up following the method proposed by Giarratana et al. [33]; a total of 189 anchovy fillets were experimentally parasitized with a total of 378 *Anisakis* larvae. Two notches of 3–4 mm in size were incised on each fillet to contain a larva (two larvae on each fillet). Then, each notch was closed by using a commercial solution of cyano acrylamide (Loctite, Italy). The encysted anchovy fillets were marinated for 24 h at $4 \pm 1\ ^\circ\text{C}$ in 300 mL of the typical industrial marinating solution (IMS): 1:1 (*vol/vol*) of distilled water and vinegar (60 g/L of acetic acid), 30 g/L of NaCl, and 10 g/L of citric acid. After the marinating process, each fillet was covered with 7.5 mL of each FOO tested and maintained at $20 \pm 1\ ^\circ\text{C}$. The larvae viability was checked starting from the second day of exposure. Encysted fillets submerged in 7.5 mL of VOO were used as CTL. For each oil at each time interval, the viability of 6 larvae (3 fillets) was checked. If all 6 larvae were found to be dead, the viability evaluation was continued until a living *Anisakis* larvae was found [15]. All the experiments were carried out three times in separate conditions and at different times.

### 2.5. Microbiological Analysis of Marinated Anchovy Fillets

For the microbiological analysis, 72 anchovy fillets (not experimentally parasitized with *Anisakis* larvae) for each of the three replications (total of 216 fillets) were marinated for 24 h at $4 \pm 1\ ^\circ\text{C}$ in the typical IMS. After the marinating process, each fillet was covered with 7.5 mL of both Mix-FOO and Cm-FOO, while fillets submerged in 7.5 mL of VOO were used as CTL. The obtained samples were kept at $20 \pm 1\ ^\circ\text{C}$ for 14 days and analyzed periodically.

#### Microbiological Analysis

At sampling time, 3 samples (each consisting of 2 fillets) were processed for each treatment (Mix-FOO, Cm-FOO, and CTL). The sample (about 20–25 g), transferred into a stomacher bag with the addition of peptone water (0.1% of peptone) in a ratio of 1:9 (*w/v*) at room temperature, was homogenized for 60 s at 230 rpm with a stomacher (Stomacher® 400 Circulator, International PBI S.p.A., Milan, Italy), and tenfold dilutions were prepared in peptone water (0.1%). An aliquot of 1 mL was plated, in duplicate, both on Violet Red Bile Glucose Agar (Biolife, Milan, Italy) for the count of *Enterobacteriaceae* after one day of incubation at 37 °C according to ISO 21528-2:2017 [34] and on Lyngby Iron Agar (Oxoid LTD., Basingstoke, Hampshire, UK) for the count of specific spoilage organisms (SSOs) after 3 days of incubation at 20 °C [35]. Among SSOs, nonsulphide-producing bacteria and sulphide-producing bacteria were separately counted. Black colonies were recorded as sulphide producers, whereas white colonies were counted as sulphide nonproducers. A representative percentage (about 10%) of white and black colonies was chosen and purified by restreaking onto tubes of Trypticase soy agar (Oxoid, Basingstoke, UK). These isolates were identified with MALDI-time-of-flight MS (MALDI-TOF MS) methods. A Vitek MS

Axima Assurance mass spectrometer (bioMerieux, Firenze, Italy) was used in positive linear mode with a laser frequency of 50 Hz, an acceleration voltage of 20 kV, and an extraction delay time of 200 ns. The mass spectra range was set to detect from 2,000 to 20,000 Da. MALDI-TOF generated unique MS spectra for each tested colony that were transferred into the SARAMIS software (Spectral Archive and Microbial Identification System, database version V4.12, software year 2013, bioMerieux, Firenze, Italy) and compared to the database of reference bacteria spectra and super spectra, obtaining the identification at the genus and species levels. Microbiological data were expressed as logarithms of the number of colony-forming units per gram of samples (Log CFU/g) and reported in a table as main value $\pm$ standard deviation.

### 2.6. Sensory Evaluation of Marinated Anchovy Fillets

For the sensory evaluation, during the third replication for the microbiological analysis, an additional 108 fillets (not experimentally parasitized with *Anisakis* larvae) were marinated as previously described, and the analysis was performed at 0, 7, and 14 days of storage. For each treatment (Mix-FOO, Cm-FOO, and CTL) and time interval, 2 fillets were used for the e-sensing analysis and 10 for traditional sensory acceptability.

### 2.6.1. E-Sensing Analysis

E-tongue: a potentiometric e-tongue ($\alpha$Astree, Alpha M.O.S., Toulouse, France), equipped with seven sensors specifically designed for food and beverage analysis, an Ag/AgCl reference electrode (Metrohm, Singapore Pte Ltd., Singapore), a mechanical stirrer, a 48-position auto-sampler, and an electronic unit for signal amplification and analog to digital conversion, was employed. The assays were carried out using sample aqueous solutions, prepared as follows: the anchovy fillet was homogenized with 50 mL of bidistilled water using an Ultra-Turrax homogenizer (T25, Ika Works Inc., Wilmington, NC, USA). The solution was centrifuged at $3000 \times g$ for 20 min, filtered, and used for further analysis. Prior to each analysis, the sensors were conditioned and calibrated with 0.01 M hydrochloric acid. Each measurement lasted 120 s, and the sensors were rinsed for 10 s with bidistilled water before each analysis. Data taken as the average of the last 10 s were used for further statistical analysis. Moreover, each sample was tested 10 times, and the first 6 measurements were discarded, in order to obtain the most stable possible potentiometric signals.

E-nose: the device used in this study (FOX 4000, Alpha M.O.S., Toulouse, France) is equipped with an array of 18 MOS (metal–oxide semiconductor) gas sensors, whose resistance is modulated in the presence of a target gas or vapor, combined with an automatic headspace sampler HS100. To perform the analysis, 1 g of each sample was weighed into a septa-sealed screw cap bottle and positioned in the autosampler racket. The instrumental settings used to perform the analysis were as shown (Table 1).

**Table 1.** E-nose settings used to perform the sensory analysis on the marinated anchovy fillets.

| Acquisition | Oven | Injection | Syringe | Agitator |
|---|---|---|---|---|
| Duration 120 s | Incubation time 600 s | Volume 1000 μL | Flushing time 120 s | Speed 500 rpm |
| Period 1 s | Incubation temperature 40 °C | Speed 1000 μL/s | Temperature 50 °C | On 5 s |
| Time 1080 s | | | Fill speed 500 μL/s | Off 2 s |
| Carrier gas flow 150 mL/min | | | | |

E-eye: a computer vision system (CVS), Iris Visual Analyzer 400 (Alpha M.O.S., Toulouse, France), was used to assess the color of the different anchovy fillets. The instrument used in this study consists of a closable measurement chamber of large dimensions, which guarantees controlled light conditions, and a CCD (charge-coupled device) camera with 16 million colors for high-resolution data acquisition. To perform the analysis, each sample was positioned in the available surface of the measurement chamber. Then, a picture was taken with a white background and top lighting only. Finally, the RGB code

corresponding to each color extracted from the different images was obtained and used as input for the statistical analysis.

Data processing: exploratory data analysis was performed with principal component analysis (PCA), using the native instrument software (AlphaSoft v14.1).

### 2.6.2. Sensory Acceptability

Sensory acceptability was determined using a panel of 20. Fourteen males and six females, ranging from 20 to 60 years old, were recruited from the Department of Veterinary Sciences (University of Messina, Italy) to take part in the sensory analysis. The criterion for the selection of panelists was the experience in evaluating seafood [36]. Six marinated anchovy fillets per group of samples were served to the panelists at room temperature ($20 \pm 3$ °C) in white porcelain trays coded with random three-digit numbers. The panelists evaluated the samples for odor, appearance, taste, and overall acceptance on a 9-point hedonic scale (0 = poor, 9 = excellent) [37].

### 2.7. Statistical Analysis

Obtained data were tested by the Kolmogorov–Smirnov method to evaluate their normal distribution, and if necessary, they were properly normalized. The one-way ANOVA was used to assess whether any differences observed in the results of each analysis were statistically significant. The post hoc Tukey's honest significant difference procedure was performed for the multiple comparisons within the obtained ANOVA data. The critical significance level (p) was set at 5% (0.05), and all tests were performed two sided. The descriptive statistics were carried out using Microsoft Excel (version 2019), while all the other statistical analyses were performed using IBM SPSS statistical software (version 24.0).

### 3. Results and Discussion

#### 3.1. Chemical Composition of the Olive Oils

In order to assess the effect of spices, herbs, and plants on the quality of olive oil, free acidity (FA), peroxide value (PV), and the specific coefficients of extinction at 232 and 270 nm ($K_{232}$ and $K_{270}$) were determined. Concerning FA, the addition of Mix and Cm on olive oil showed no negative effect, and the values were comparable to those of the VOO (Table 2) ($p \geq 0.05$). Some studies displayed that the addition of spices did not influence the FA; for example, Sousa et al. [38] showed not significant increases in the FA value of oils with hot chili peppers, laurel, oregano, and pepper, relative to the control olive oil. The PV indicates the formation of primary compounds of oxidation. The addition of the combination of parsley, fresh garlic, and lemon (Mix) induced a significant decrease of PV compared with VOO ($p \leq 0.05$), while Cm-FOO displayed little but significant augmentation compared with VOO ($p \leq 0.05$). According to Sousa et al. [38], the addition of spices can exhibit positive or negative effects on oil PV values depending on the nature of the spices. The FOOs reported higher values of specific extinction at 270 than the CTL oil, while at 232, no significant differences were observed between Mix-FOO and CTL ($p \geq 0.05$), whose values were significantly lower than Cm-FOO ($p \leq 0.05$). The presence of some compounds in spices, herbs, and plants that can absorb 270 nm significantly increase these coefficients. As reported by Gambacorta et al. [9], the addition of hot pepper, garlic, oregano, and rosemary increased the values of $K_{270}$.

**Table 2.** Effect of adding spices on olive oil quality parameters (free acidity, peroxide value, specific extinction), total biophenol contents, and radical scavenging activity (DPPH μL/mL).

| Parameters | Mix-FOO [1] | Cm-FOO [2] | CTL [3] |
|---|---|---|---|
| Free Acidity (% C18:1) | 0.3 ± 0.032 | 0.39 ± 0.000 | 0.36 ± 0.032 |
| PV (meq of $O_2$/Kg) | 3.5 ± 0.000 [a] | 7.33 ± 1.041 [b] | 6.33 ± 0.289 [c] |
| $K_{232}$ | 2.12 ± 0.032 [a] | 3.31 ± 0.075 [b] | 2.06 ± 0.196 [a] |
| $K_{270}$ | 0.23 ± 0.009 [a] | 0.80 ± 0.049 [b] | 0.12 ± 0.024 [c] |
| Bio-phenols (mg/Kg) | 740.4 ± 2.76 [a] | 734.1 ± 3.46 [a] | 1067.1 ± 5.05 [b] |
| DPPH test ($IC_{50}$ in μL/mL) | 13 ± 1.73 [a] | 21.17 ± 0.5 [b] | 18.83 ± 6.71 [ab] |

Values are reported as mean ± standard deviation of 3 replicants. Different letters in the same row represent significant differences at $p \leq 0.05$. [1] Virgin olive oil flavored with parsley, fresh garlic, and lemon; [2] virgin olive oil flavored with cumin; [3] virgin olive oil.

The total of biophenols was significantly higher in the VOO than in the FOOs ($p \leq 0.05$), and they varied in the following order: VOO (1067.1 mg/kg) > Mix-FOO (740.4 mg/kg) > Cm-FOO (734.1 mg/kg). The same observation has been found by Baiano et al. [39], which showed that VOO is richer in biophenols than FOOs. They explained the differences between the phenolic contents of unflavored and flavored oils by pointing to the interactions that take place between the olives and the flavoring agents during the extraction phase as being responsible for the formation of bonds between phenolics and the components of the spices, herbs, and fruits. The radical scavenging activity of Cm-FOO and VOO are comparable ($p \geq 0.05$), while the incorporation of the parsley, fresh garlic, and lemon in oil has a positive effect on radical scavenging activities. The antioxidant compounds present in the herbs and spices differ significantly, as do their antioxidant potential, which may influence the results obtained. Thus, Sousa et al. [38] strengthens the hypothesis that other compounds different from phenolics and/or synergic reactions could play an important function in the antioxidant properties of olive oil.

The fatty acids composition was assessed in the tested olive oils; their details are reported in Table 3. In all samples, oleic acid (C18:1) was the most abundant fatty acid, followed by linoleic acid (C18:2) and palmitic acid (C16:0). This is considered a typical fatty acid profile of extra virgin olive oil for the monovarietal Chétoui characterized by an acceptable monounsaturated fatty acids composition, which is very important due its nutritional implication and effect on the oxidative stability of oils [40]. On the whole, the aromatization of the olive oil did not significantly influence the amounts of individual fatty acids, with the exception of linoleic acid (C18:2), which increased with the addition of flavoring agents ($p \leq 0.05$), and palmitoleic acid (C16:1), which was significantly higher in the Cm-FOO ($p \leq 0.05$).

**Table 3.** Fatty acids profile (%) of different flavored virgin olive oils.

| Parameters | Fatty Acids | Mix-FOO [1] | Cm-FOO [2] | CTL [3] |
|---|---|---|---|---|
| Saturated fatty acids | C 16:0 [4] | 11.98 ± 0.50 | 12.42 ± 0.50 | 12.7 ± 0.49 |
| | C 18:0 [5] | 2.95 ± 0.24 | 3 ± 0.25 | 3.08 ± 0.25 |
| | C 20:0 [6] | 0.65 ± 0.15 | 0.64 ± 0.15 | 0.61 ± 0.15 |
| Monounsaturated fatty acids | C 16:1 [7] | 0.33 ± 0.2 [a] | 0.5 ± 0.20 [b] | 0.25 ± 0.19 [a] |
| | C 18:1 [8] | 63.46 ± 1.57 | 61.89 ± 1.42 | 63.84 ± 1.58 |
| | C 20:1 [9] | 0.4 ± 0.17 | 0.41 ± 0.17 | 0.39 ± 0.17 |
| Poly-unsaturated fatty acids | C 18:2 [10] | 17.24 ± 1.30 [a] | 17.62 ± 1.28 [a] | 15.52 ± 1.29 [b] |
| | C 18:3 [11] | 0.47 ± 0.16 | 0.47 ± 0.16 | 0.47 ± 0.16 |

Values are reported as mean ± standard deviation of 3 replicants. Different letters in the same row represent significant differences at $p \leq 0.05$. [1] Virgin olive oil flavored with parsley, fresh garlic, and lemon; [2] virgin olive oil flavored with cumin; [3] virgin olive oil; [4] palmitic acid; [5] stearic acid; [6] arachic acid, [7] palmitoleic acid; [8] oleic acid; [9] gondoic acid; [10] linoleic acid; [11] linolenic acid.

Despite the weak variations observed, the results obtained from the flavored oils are still in accordance with the maximum permitted levels in order to be considered extra virgin olive oils [26].

### 3.2. Anisakicidal Activity

3.2.1. In Vitro Anisakicidal Activity

The normalized mean viability scores of *Anisakis* larvae treated with Mix-FOO and Cm-FOO are reported in Figure 1. In both FOOs, the total inactivation of the larvae occurred within 24 h of exposure. The Mix-FOO showed the greatest effectiveness with a LT50 and LT100 of 16.2 h and 22 h, respectively, while for Cm-FOO, the LT50 was 18 h and the LT100 was 24 h. Overall, the normalized viability scores showed that Mix-FOO was the most effective against *Anisakis* larvae. The results of statistical analysis confirmed the evident significant differences between both FOOs and the CTL samples ($p \leq 0.05$), while no significant differences were observed between the Mix-FOO and Cm-FOO for all the intervals ($p \geq 0.05$). The larvae exposure to VOO in the CTL samples allowed us to consider the results with confidence, since no larvae mortality was detected but only a minimal viability reduction starting at the eighteenth hour.

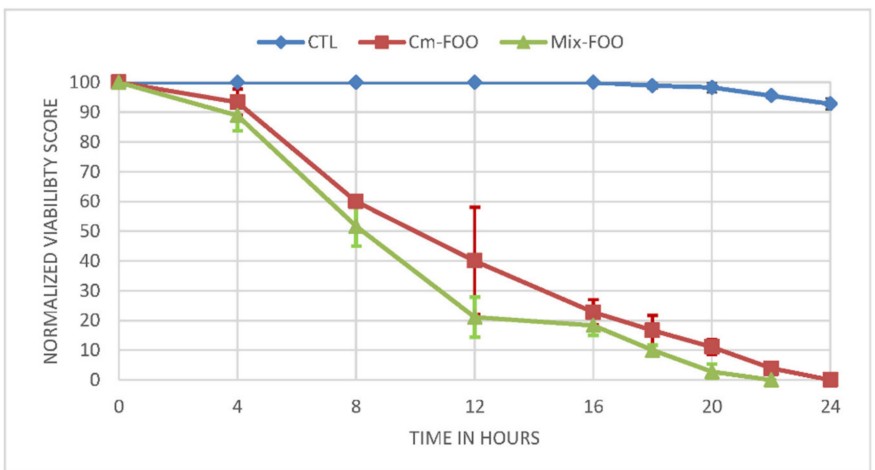

**Figure 1.** Normalized mean viability (%) of *Anisakis* larvae type I in virgin olive oil flavored with cumin (Cm-FOO); virgin olive oil flavored with parsley, fresh garlic, and lemon (Mix-FOO); and virgin olive oil (CTL).

The efficacy of the Mix-FOO could be related to the synergic effect of the different active compounds of parsley, fresh garlic, and lemon, which together result in higher anisakicidal activity. In this regard, there is much evidence that the beneficial effects of different natural compounds may be greater when combined with one another [41]. On the other hand, it cannot be ruled out that a key role might be played by one or only some of the active compounds in the flavored oil. Apigenin, aspirin, and 6"-acetylapiin in parsley [42]; allyl polysulfides in garlic [43]; and limonene in lemon [44] have been reported among the main active compounds, respectively, and they could be responsible for the anisakicidal effect.

The in vitro anisakicidal efficacy of the Cm-FOO at 10% was higher than that observed in our previous preliminary in vitro study, where flavored olive oil with cumin at 5% was tested against *Anisakis* larvae [10]. In fact, at 5% cumin concentration, complete inactivation of the *Anisakis* larvae was observed after 186 h (7.7 days) of exposure, far greater than the 24 h observed in the present study with 10% cumin. This difference is probably related to the greater content of active constituents, such as β-pinene, γ-terpinene, p-cymene, and cumin aldehyde, which are the most representative volatile compounds of the cumin flavored olive oil obtained by cold pressing [45].

Both the Mix-FOO and Cm-FOO determined different types of lesions in all dead parasites, which were similar to those observed in *Anisakis* larvae treated with flavored olive oils of cinnamon, rosemary, cardamom, and ginger [15]. In particular, more or less severe damage occurred in all the larvae, which included continuous lesions of the cuticle and interruptions of the digestive tract sometimes associated with its leakage through cuticle lesions (Figure 2a–c).

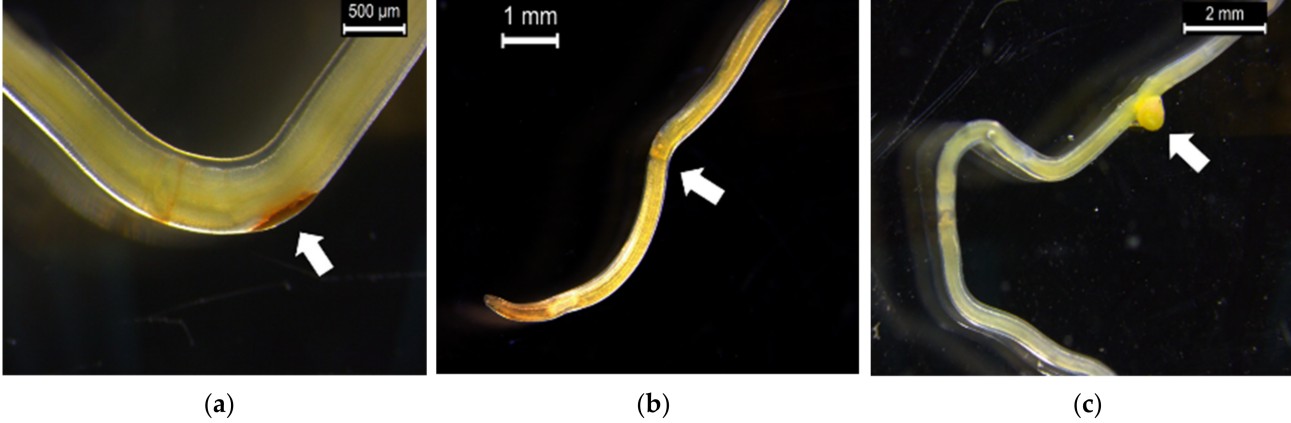

| (a) | (b) | (c) |

**Figure 2.** Examples of lesions observed under stereoscopic microscope in *Anisakis* larvae treated both with Mix-FOO (virgin olive oil flavored with parsley, fresh garlic, and lemon) and with CM-FOO (virgin olive oil flavored with cumin): (**a**) continuous lesions of the cuticle; (**b**) interruptions of the digestive tract; (**c**) continuous lesions of the cuticle associated with leak of the digestive tract.

### 3.2.2. Ex Vivo Anisakicidal Activity

The Mix-FOO and Cm-FOO determined a complete inactivation of the *Anisakis* larvae experimentally parasitized in the marinated anchovy fillets (Table 4). Unlike what was observed in the in vitro experiment, the Cm-FOO had a greater anisakicidal activity than that observed for the Mix-FOO. In particular, the total larvae inactivation occurred after the 8th day for the Cm-FOO and after 10th day for the Mix-FOO. This difference could be related to the chemical composition of the cumin, which allows for an easier diffusion of the oil through the tissues of the fish. The fish tissues protect the larvae, which are inactivated after more time than in the in vitro experiment. In a preliminary ex vivo study, the Cm-FOO at 5% completely inactivated the *Anisakis* larvae after 5 days, showing a greater anisakicidal activity than that herein observed, despite the fact that a greater concentration of cumin (10%) was used [10]. This discrepancy is probably related to the different chemical compositions of the olive oils. In fact, although the olive oils were both obtained through the same process of extraction and from the same cultivar of olive fruits (Chétoui) picked in the region of Takelsa, several environmental factors (such as temperature, atmospheric precipitation, and soil modifications) could have affected the chemical composition of the oils and their relative "richness" in bioactive compounds. Furthermore, in this experiment, lesions were observed in the *Anisakis* larvae, which, nevertheless, were less severe than those observed in vitro, limited to continuous lesions of the cuticula and the interruption of digestive tract (Figure 2a,b).

**Table 4.** Number of individuals and relative viability score over time of *Anisakis* larvae experimentally parasitized in marinated anchovy fillets exposed to different flavored virgin olive oils.

| Viability Score | 2nd Day | | | 4th Day | | | 6th Day | | | 8th Day | | | 10th Day | | | 12th Day | | | 14th Day | | |
|---|---|---|---|---|---|---|---|---|---|---|---|---|---|---|---|---|---|---|---|---|---|
| | r.1 | r.2 | r.3 | r.1 | r.2 | r.3 | r.1 | r.2 | r.3 | r.1 | r.2 | r.3 | r.1 | r.2 | r.3 | r.1 | r.2 | r.3 | r.1 | r.2 | r.3 |
| CTL [1]    3 | 6 | 5 | 6 | 5 | 5 | 4 | 5 | 4 | 4 | 4 | 4 | 4 | 5 | 4 | 3 | 2 | 4 | 3 | - | - | - |
| 2 | - | 1 | - | 1 | 1 | 2 | 1 | 2 | 2 | 2 | 2 | 2 | 1 | 2 | 3 | 4 | 2 | 3 | 3 | 4 | 5 |
| 1 | - | - | - | - | - | - | - | - | - | - | - | - | - | - | - | - | - | - | 3 | 2 | 1 |
| 0 | - | - | - | - | - | - | - | - | - | - | - | - | - | - | - | - | - | - | - | - | - |
| Mix-FOO [2]    3 | 4 | 4 | 4 | - | - | - | - | - | - | - | - | - | - | - | - | - | - | - | - | - | - |
| 2 | 2 | 2 | 2 | 4 | 3 | 4 | - | - | - | - | - | - | - | - | - | - | - | - | - | - | - |
| 1 | - | - | - | 2 | 3 | 2 | 4 | 3 | 4 | 1 | 2 | 2 | - | - | - | - | - | - | - | - | - |
| 0 | - | - | - | - | - | - | 2 | 3 | 2 | 5 | 4 | 4 | 18 | 18 | 18 | - | - | - | - | - | - |
| Cm-FOO [3]    3 | 3 | 2 | 4 | - | - | - | - | - | - | - | - | - | - | - | - | - | - | - | - | - | - |
| 2 | 3 | 4 | 2 | - | 1 | 2 | - | - | - | - | - | - | - | - | - | - | - | - | - | - | - |
| 1 | - | - | - | 5 | 4 | 3 | 2 | 1 | 2 | - | - | - | - | - | - | - | - | - | - | - | - |
| 0 | - | - | - | 1 | 1 | 1 | 4 | 5 | 4 | 24 | 24 | 24 | - | - | - | - | - | - | - | - | - |

r. = replicate. [1] Virgin olive oil; [2] virgin olive oil flavored with parsley, fresh garlic, and lemon; [3] virgin olive oil flavored with cumin.

### 3.3. Microbiological Analysis

The microbiological results obtained for the marinated anchovy fillets treated with the FOOs are presented in Figure 3. Both the FOOs showed antibacterial effects against spoilage bacteria, considering the lower loads observed in the treated samples compared to the CTL ones. No *Enterobacteriaceae* were detected until the end of the study (after 14 days of treatment) for either the Mix-FOO, Cm-FOO, or CTL, probably due to the antimicrobial effects of the marinating treatment. In fact, the acid environment (pH $4 \pm 0.25$) determined by the IMS was not favorable to the growth of *Enterobacteriaceae*. After one day of treatment, no SSO growth was observed in samples exposed to both FOOs, while a rather low count of white colonies (<Log 1 CFU/g) was detected in the CTL samples ($p \geq 0.05$). After one week of treatment, Mix-FOO and Cm-FOO efficiently inhibited the growth of spoilage bacteria, never exceeding log 3 CFU/g and log 2 CFU/g, lower than in the CTL samples ($p \leq 0.05$). The value of 7 log CFU/g of SSOs, considered to be the upper microbiological limit of the acceptable quality of food [46,47], was reached on the 14th day for the CTL samples, while counts < 5 log CFU/g were observed in the samples treated with both FOOs ($p \leq 0.05$). No substantial differences were noticed between the antibacterial activity of Mix-FOO and Cm-FOO ($p \geq 0.05$). The counts of the white colonies were always greater than those of the black colonies in all the samples, regardless of treatment with FOOs. Only on the 14th day were low counts of black colonies detected in the fillets treated with both FOOs.

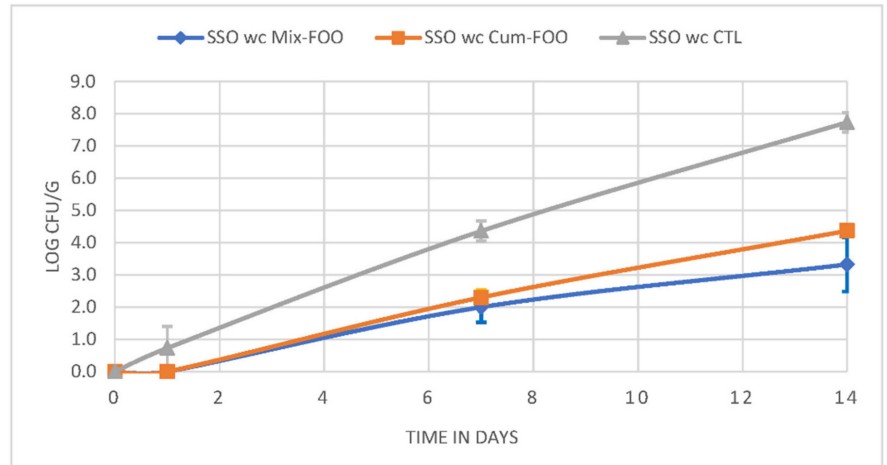

**Figure 3.** Microbial load of the white colonies of specific spoilage organisms in Mix-FOO (virgin olive oil flavored with parsley, fresh garlic, and lemon), Cm-FOO (virgin olive oil flavored with cumin), and CTL (virgin olive oil).

Considering the SSO's population, the MALDI-TOFF analysis allows us to identify the strains as belonging to genera *Pseudomonas*, *Lactobacillus*, and *Shewanella*. In particular, among the white colonies, *Pseudomonas fragi*, *Pseudomonas putida*, *Pseudomonas fluorescens*, and *Lactobacillus alimentarius* were identified, whereas the few isolated black colonies were all *Shewanella putrefaciens*. These isolations agree with the results of several other studies concerning the spoilage bacteria of marinated fish [48,49]. The results of this study confirm such reports by other authors regarding the antimicrobial effects of the lemon, parsley, garlic, and cumin [50]. In fact, lemon, parsley, garlic, and cumin have already been efficiently used to extend the shelf life of fishery products (gilthead breams, silver carp, herring, and rainbow trout), thanks to the antimicrobial effects of their bioactive compounds, without particular restrictions related to sensory influences, since they are widely appreciated by consumers [51–54].

### 3.4. Sensory Evaluation of Marinated Anchovy Fillets

#### 3.4.1. E-Sensing Evaluation

The plots of the first two principal components of the PCA models built with the electronic tongue (Figure 4), electronic nose (Figure 5), and electronic eye are as shown (Figure 6). Data related to the marinated anchovy fillets treated with FOOs at 0, 7, and 14 days of storage show the taste, odor, and color evolution for each group during that time. In all three plots, the first component better explains the difference between groups, while along the second component, the variance due to storage time seems clearer. The e-tongue analysis showed that the greatest change in taste profile during storage was observed in the CTL group, while the two Mix-FOO and Cm-FOO groups had less variation over time. The odor profile detected from e-nose sensors changed during storage from 0 to 14 days, with a similar trend in all three groups, and shows variability between samples from the same group on the same date. With regards to the color profile of marinated anchovy fillets, the Cm-FOO group showed the highest stability over time.

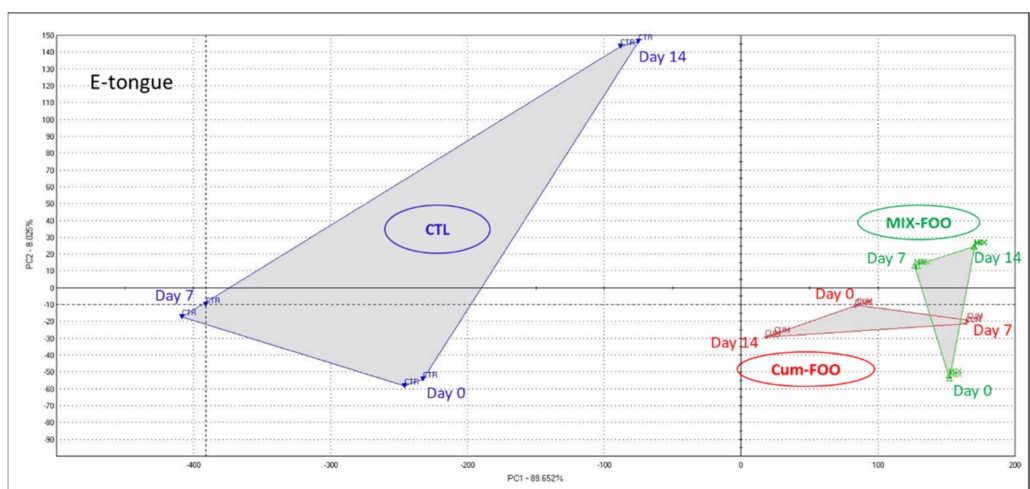

**Figure 4.** E-tongue: Taste map (principal component analysis) of the 3 groups of anchovy fillets [CTL (virgin olive oil), Mix-FOO (virgin olive oil flavored with parsley, fresh garlic, and lemon), and Cm-FOO (virgin olive oil flavored with cumin)] during the trial (day 0, 7, and 14).

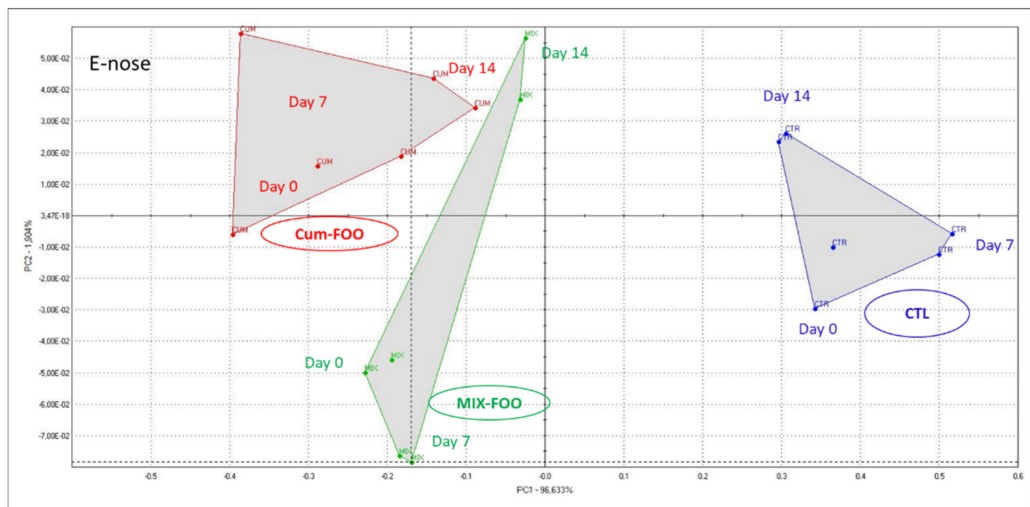

**Figure 5.** E-nose: Odor map (principal component analysis) of the 3 groups of anchovy fillets [CTL (virgin olive oil), Mix-FOO (virgin olive oil flavored with parsley, fresh garlic, and lemon), and Cm-FOO (virgin olive oil flavored with cumin)] during the trial (day 0, 7, 14).

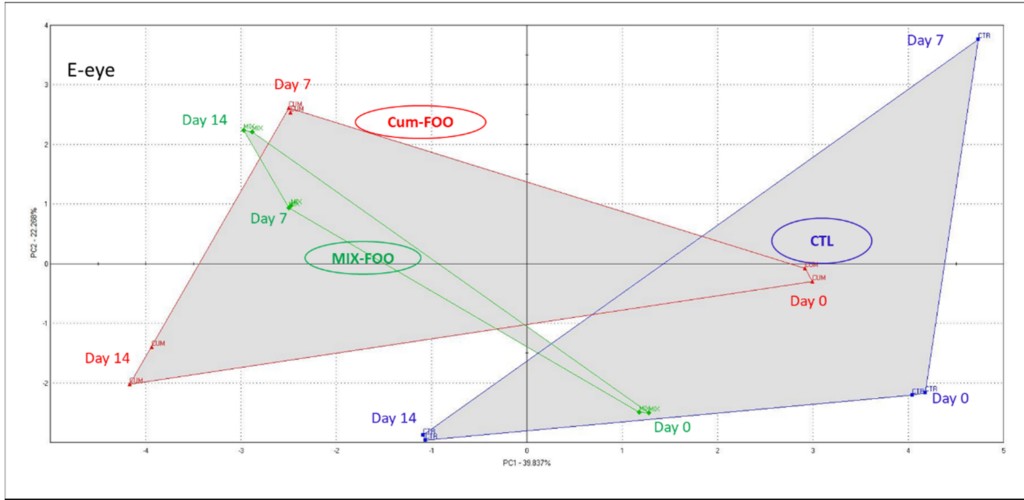

**Figure 6.** E-eye: Color map (principal component analysis) of the 3 groups of anchovy fillets [CTL (virgin olive oil), Mix-FOO (virgin olive oil flavored with parsley, fresh garlic, and lemon), and Cm-FOO (virgin olive oil flavored with cumin)] during the trial (day 0, 7, 14).

Overall, during storage, the two Mix-FOO and Cm-FOO groups showed a similar trend to the CTL group, proving to be stable over time. Considering the low influence in sensory evaluation, the enriched oil mixtures can represent an effective alternative method to the freezing process required by European Regulation EC No 853/2004.

### 3.4.2. Sensory Acceptability

The sensory evaluation of marinated anchovy fillets with FOOs appears to be necessary, since the increased shelf life of these fish products is ineffective if sensory properties are unacceptable to consumers. Thus, sensory analysis was conducted to detect the acceptability of FOO groups. The acceptance test results are presented in Table 5.

**Table 5.** Mean sensory scores of marinated anchovy fillets exposed to different flavored virgin olive oil during storage at 4 °C.

| Anchovy Fillets Groups | | Storage (Days) | | |
|---|---|---|---|---|
| | | **0** | **7** | **14** |
| Odor | CTL [1] | 8.2 ± 0.4 | 8.0 ± 0.3 | 7.5 ± 0.5 |
| | Mix-FOO [2] | 8.6 ± 0.5 | 7.9 ± 0.2 | 7.4 ± 0.5 |
| | Cm-FOO [3] | 8.1 ± 0.3 | 8.0 ± 0.0 | 7.5 ± 0.5 |
| Color | CTL | 8.5 ± 0.5 [a] | 7.8 ± 0.4 [a] | 7.2 ± 0.4 [b] |
| | Mix-FOO | 8.4 ± 0.5 [a] | 7.9 ± 0.3 [a] | 7.4 ± 0.5 [b] |
| | Cm-FOO | 8.1 ± 0.3 | 8.0 ± 0.3 | 7.8 ± 0.4 |
| Taste | CTL | 8.4 ± 0.5 [a] | 7.5 ± 0.5 [a] | 7.1 ± 0.2 [b] |
| | Mix-FOO | 8.5 ± 0.5 | 8.2 ± 0.5 | 7.8 ± 0.4 |
| | Cm-FOO | 8.3 ± 0.5 | 8.0 ± 0.3 | 7.8 ± 0.4 |
| Overall acceptance | CTL | 8.5 ± 0.5 | 8.0 ± 0.3 | 7.5 ± 0.6 |
| | Mix-FOO | 8.6 ± 0.6 | 8.2 ± 0.5 | 7.9 ± 0.3 |
| | Cm-FOO | 8.3 ± 0.5 | 8.1 ± 0.2 | 7.9 ± 0.3 |

Different letters in the same row represent significant differences at p ≤ 0.05. [1] Virgin olive oil; [2] virgin olive oil flavored with parsley, fresh garlic, and lemon; [3] virgin olive oil flavored with cumin.

Scores that were greater than 7 throughout storage indicated good acceptability without significant differences between CTL and FOO groups. The odor, color, and taste scores showed a downward trend during storage in all groups. The color acceptance scores of the CTL and Mix-FOO anchovy fillets were significantly lower (p ≤ 0.05) at day 14; on the other hand, the Cm-FOO color score seems more stable over time. Storage time has no effect on the taste scores of the Mix-FOO and Cm-FOO groups, but the CTL group showed a lower score (p ≤ 0.05) at day 14. These results confirmed the e-sensing analysis and the potential use of FOOs for improving shelf life of marinated anchovies.

## 4. Conclusions

The herbs, spices, and plants used for the preparation of Mix-FOO and Cm-FOO are widely used as natural additives to enrich the sensory quality of foods, especially of fishery products. The results obtained from the analysis of the antioxidant potential and antimicrobial and anisakicidal activity are promising for the possible use Mix-FOO and Cm-FOO to extend the shelf life of marinated anchovy fillets and to manage the *Anisakis* risk at industrial levels. Furthermore, considering their appreciated sensory attributes, their use is not conditioned by their influence on the organoleptic properties of the food products. The use of FOO in foods could represent a sustainable approach to ensure food safety and quality consistent with the guidelines proposed by the FAO [4]. In fact, they contribute to the protection of biodiversity and the optimization of human and natural resources, in addition to being culturally acceptable, economically fair, accessible, and nutritionally healthy.

**Author Contributions:** Conceptualization, N.T., S.N.-M., R.M., and A.G.; methodology, F.G., N.T., and A.R.D.R.; software, A.R.D.R.; validation, N.T., F.G., and L.N.; formal analysis, N.T., A.E.-D., L.N., and A.R.D.R.; data curation, N.T., F.G., and A.R.D.R.; writing—original draft preparation, L.N., A.R.D.R., and N.T.; writing—review and editing, L.N., A.R.D.R., F.G., and N.T.; supervision, R.M. and A.G. All authors have read and agreed to the published version of the manuscript.

**Funding:** This research received no external funding.

**Institutional Review Board Statement:** Not applicable.

**Informed Consent Statement:** Written informed consent was obtained from all subjects involved in the study.

**Conflicts of Interest:** The authors declare no conflict of interest.

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
