# Peer review of "Marinated Anchovies (Engraulis encrasicolus) Prepared with Flavored Olive Oils (Chétoui cv.): Anisakicidal Effect, Microbiological, and Sensory Evaluation"

_sustainability, doi:10.3390/su13095310_

Round 1

Reviewer 1 Report

Submitted manuscript reports possibility in using aromatized virgin olive oils as natural alternative to the freezing treatment in the marinating process of anchovy fillets. Although the topic is of interest, some minor and major points should be raised before accepting the paper for publication. Please find down all suggestions, doubt and questions:

Line 23. in abstract it cannot be state 'good'. Be precise and incorporate numbers, like …by 20% higher than xy …or similar

Whole last sentence is unnecessary, and results generated in the study were poorly presented, thus please rephrase whole abstract

Line 54, delete space

Line 95 delete they

Line 125, is this lab scale mill, if yes indicate clearly here

Line 137 for K number use index for 232 and 270

Reference 26 – check is this is correct – (Commission Regulation (EEC) No 2568/91)

Line 132 – How the unflavored oil were kept?

Unflavored oil is in fact virgin olive oil. Why you decide to use term unflavored? It is probably pointless since this category does not exist, please reconsider use as VOO

Line 170-174  This is direct copy from the method, please use past tense and since this is explanation of FAME preparation move and incorporate in line 163.

Line 251. The remaining 108 fillets ?? Please specify which samples were used for sensory analyses

line 255 delete In the present work,

line 262 ULTRA-TURRAX, do not use upper case letters, and add model, County etc

line 287 was used trained panelists? 20 consumers are too small number for consumer’s research

line 298 on the

line 300 rephrase flavor – you use spice/herbs …so use same phrases as in MM section

line 303 delete that

line 315 mgEAG/kgOO – the measurement unit is misspelled, GAE instead EAG, however is enough mg/kg  (in MM section is good explained and this meas. Unit is widely accepted)

Table 2, also wrong meas. Unit for TPC

Table 2. Please perform a statistical analysis, add significance and then check if what is written in the discussion is still valid

Same goes for Table 3

Table 5. are presented means values or median values?

Line 482 please p < 0.05 change as ≤ (apply in whole MS is applicable)

Author Response

Reviewer 1

Submitted manuscript reports possibility in using aromatized virgin olive oils as natural alternative to the freezing treatment in the marinating process of anchovy fillets. Although the topic is of interest, some minor and major points should be raised before accepting the paper for publication.

  1. Line 23. in abstract it cannot be state 'good'. Be precise and incorporate numbers, like …by 20% higher than xy …or similar
  2. According to the Reviewer’s suggestion, the abstract was edited.

  1. Whole last sentence is unnecessary, and results generated in the study were poorly presented, thus please rephrase whole abstract
  2. According to the Reviewer’s suggestion, the abstract was edited.

  1. Line 54, delete space
  2. According to the Reviewer’s suggestion, the manuscript was edited.

  1. Line 95 delete they
  2. According to the Reviewer’s suggestion, the manuscript was edited.

  1. Line 125, is this lab scale mill, if yes indicate clearly here
  2. According to the Reviewer’s suggestion, the manuscript was edited.

  1. Line 137 for K number use index for 232 and 270
  2. According to the Reviewer’s suggestion, the manuscript was edited.

  1. Reference 26 – check is this is correct – (Commission Regulation (EEC) No 2568/91)
  2. According to the Reviewer’s suggestion, the reference 26 was checked and edited.

  1. Line 132 – How the unflavored oil were kept?
  2. According to the Reviewer’s suggestion, the manuscript was edited and this information added.

  1. Unflavored oil is in fact virgin olive oil. Why you decide to use term unflavored? It is probably pointless since this category does not exist, please reconsider use as VOO
  2. We have used the term “Unflavored” just to explain that we don’t have add any flavoring agent. According to the Reviewer’s suggestion, the word “Unflavored” was edited and replaced with the word “VOO”.

  1. Line 170-174 This is direct copy from the method, please use past tense and since this is explanation of FAME preparation move and incorporate in line 163.

10 According to the reviewer’s suggestion, the manuscript was edited.

  1. Line 251. The remaining 108 fillets ?? Please specify which samples were used for sensory analyses
  2. According to the Reviewer’s suggestion, the manuscript was edited specifying better which samples were used.

  1. line 255 delete In the present work,
  2. According to the Reviewer’s suggestion, the manuscript was edited.

  1. line 262 ULTRA-TURRAX, do not use upper case letters, and add model, County etc
  2. According to the Reviewer’s suggestion, the manuscript was edited.

  1. line 287 was used trained panelists? 20 consumers are too small number for consumer’s research
  2. Thanks for your attention. We used trained panelist. The sentence was corrected.

  1. line 298 on the
  2. According to the Reviewer’s suggestion, the manuscript was edited.

  1. line 300 rephrase flavor – you use spice/herbs …so use same phrases as in MM section
  2. According to the Reviewer’s suggestion, the manuscript was edited.

  1. line 303 delete that
  2. According to the Reviewer’s suggestion, the manuscript was edited.

  1. line 315 mgEAG/kgOO – the measurement unit is misspelled, GAE instead EAG, however is enough mg/kg (in MM section is good explained and this meas. Unit is widely accepted)
  2. According to the Reviewer’s suggestion, the manuscript was edited and the measurement units fixed.

  1. Table 2, also wrong meas. Unit for TPC
  2. According to the Reviewer’s suggestion, the manuscript was edited and the measurement units fixed.

  1. Table 2. Please perform a statistical analysis, add significance and then check if what is written in the discussion is still valid
  2. According to the Reviewer’s suggestion, statistical analysis was performed, the significance was added and the results were checked.

  1. Same goes for Table 3
  2. According to the Reviewer’s suggestion, statistical analysis was performed, the significance was added and the results were checked.

22.Table 5. are presented means values or median values?

  1. Data are presented as means and, thanks to your kind suggestion, it has been better specified in the table.

  1. Line 482 please p < 0.05 change as ≤ (apply in whole MS is applicable)
  2. According to the Reviewer’s suggestion, the manuscript was edited and the change was adopted throughout the text.

Reviewer 2 Report

The manuscript reports an interesting research about the solution of a real problem of foods based on marinated crude fish. The research have used state-of-the-art analytical methodologies of quality evaluation. I evidence a lack of statistical analysis of data as I reported in my notes below.

Title. I suggest changing title. The Chétoui cultivar is important but if authors report a formula as “…  of Flavored Olive (Olea europaea L.) Oils (Chétoui cv.) …” probably the manuscript will receive more attention. Authors could evaluate my proposal.

Material and methods

Line 147, 164. The complete identification of the instruments should be reported.

Line 148. Total phenolic content was obtained by mean of a calibration curve. Even if it is well known and implied, authors should define it.

Lines 168-169. Authors should better describe the injection mode of the samples. Were samples injected as hexane solution of methyl esters? What they meant as “the injection time is carried out during 27 min”? What injection mode was applied?

Lines 215-216. It is not clear the enumeration of viable larvae. Why did authors check fillets until they found a viable larva? Could they better explain the procedure?

Line 233. Authors should report the complete definition of abbreviations used. They should report the SSO complete name. I think they referred to Specific Spoilage Organism, it is well known but it is correct specify it the first time the abbreviation is reported.

Lines 288-295. Sensory acceptability. I hope panelist have tasted anchovy fillets without Anisakis.

Results and Discussion

Table 2 and 3, Figure 1 and 3. Authors should carry out statistical analyses to verify the differences among theses. An ANOVA and a post-hoc test such as Duncans’ or Tukey could be applied. Currently, it is not clear if there were differences among samples. After the statistical evaluation, authors should verify the discussion carried out at lines 298-342. Also the affirmation about differences observed between the two FOO should revised after statistical evaluations, see lines 350-352 and 362. Statistical differences of data showed on figures could be reported along text (if on the graph is difficult to evidence) or on the graphs themselves.

Lines 334-336. The authors’ statement is not completely correct. The amount of oleic acid (the most important monounsaturated fatty acid of olive oil) was not high in the oils studied. The amount lower than 70% of the total is not considered high. See the interesting review by Jimenez-Lopez et al,   Foods 2020, 9, 1014; doi:10.3390/foods9081014, and the references reported in it. Authors should modify the phrase.

Lines 449-474. E-sensing evaluation. Authors could improve the discussion for odor map, where the PC2 explained only 1,9% of the variability. Could PC1 alone explain the variability? From PCA maps are not clear the changes during storage time. Could authors give more information about these trends? Authors could evaluate my suggestion.

Author Response

Reviewer 2

The manuscript reports an interesting research about the solution of a real problem of foods based on marinated crude fish. The research have used state-of-the-art analytical methodologies of quality evaluation. I evidence a lack of statistical analysis of data as I reported in my notes below.

  1. Title. I suggest changing title. The Chétoui cultivar is important but if authors report a formula as “… of Flavored Olive (Olea europaea L.) Oils (Chétoui cv.) …” probably the manuscript will receive more attention. Authors could evaluate my proposal.
  2. According to the Reviewer’s suggestion, the title of the manuscript was edited.

Material and methods

  1. Line 147, 164. The complete identification of the instruments should be reported.
  2. According to the Reviewer’s suggestion, the manuscript was edited and the all the requested information added.

  1. Line 148. Total phenolic content was obtained by mean of a calibration curve. Even if it is well known and implied, authors should define it.
  2. According to the Reviewer’s suggestion, the manuscript was edited and the all the requested information added.

  1. Lines 168-169. Authors should better describe the injection mode of the samples. Were samples injected as hexane solution of methyl esters? What they meant as “the injection time is carried out during 27 min”? What injection mode was applied?
  2. According to the Reviewer’s suggestion, the manuscript was edited and the all the requested information added.

  1. Lines 215-216. It is not clear the enumeration of viable larvae. Why did authors check fillets until they found a viable larva? Could they better explain the procedure?
  2. According to the Reviewer’s suggestion, the manuscript was edited and the explanation of the relative procedure was improved.

  1. Line 233. Authors should report the complete definition of abbreviations used. They should report the SSO complete name. I think they referred to Specific Spoilage Organism, it is well known but it is correct specify it the first time the abbreviation is reported.
  2. Thanks to the Reviewer’s suggestion, the manuscript was edited.

  1. Lines 288-295. Sensory acceptability. I hope panelist have tasted anchovy fillets without Anisakis.
  2. Yes, panelist tested only anchovy fillets without Anisakis. The manuscript was edited and this concept was specified.

Results and Discussion

  1. Table 2 and 3, Figure 1 and 3. Authors should carry out statistical analyses to verify the differences among theses. An ANOVA and a post-hoc test such as Duncans’ or Tukey could be applied. Currently, it is not clear if there were differences among samples. After the statistical evaluation, authors should verify the discussion carried out at lines 298-342. Also the affirmation about differences observed between the two FOO should revised after statistical evaluations, see lines 350-352 and 362. Statistical differences of data showed on figures could be reported along text (if on the graph is difficult to evidence) or on the graphs themselves.
  2. According to the Reviewer’s suggestion, statistical analysis was performed, the significance was added and the results were checked.

  1. Lines 334-336. The authors’ statement is not completely correct. The amount of oleic acid (the most important monounsaturated fatty acid of olive oil) was not high in the oils studied. The amount lower than 70% of the total is not considered high. See the interesting review by Jimenez-Lopez et al, Foods 2020, 9, 1014; doi:10.3390/foods9081014, and the references reported in it. Authors should modify the phrase.
  2. Thanks and according to the Reviewer’s suggestion, the manuscript was edited modifying the sentence.

  1. Lines 449-474. E-sensing evaluation. Authors could improve the discussion for odor map, where the PC2 explained only 1,9% of the variability. Could PC1 alone explain the variability? From PCA maps are not clear the changes during storage time. Could authors give more information about these trends? Authors could evaluate my suggestion.
  2. Thanks for your kind comment. We improved the PCA maps including the storage time (Day 0, Day 7 and Day 14) in the plots. We also added some sentence in the text. We hope that E-sensing discussion has improved.

Round 2

Reviewer 1 Report

The authos have adrresed all my queastion,

after some more minor changes I suugests MS for publication.

Please, (Olea europaea L.) in Article title move to key words,

Line 364 – you have difference for C 16:1 (Cm-FOO), thus include as exception

Also, as tables should be self-explicable add explanations of abbreviations, for fatty acids and used treatments – as for Table 3, same apply in other Tables/figures

Also avoid using abbreviations in headings and subheadings.

Author Response

Reviewer 1

  1. Please, (Olea europaea L.) in Article title move to key words,
  2. According to the Reviewer’s suggestion, title and key words were edited.

  1. Line 364 – you have difference for C 16:1 (Cm-FOO), thus include as exception
  2. According to the Reviewer’s suggestion, manuscript was edited and the difference included in the text.

  1. Also, as tables should be self-explicable add explanations of abbreviations, for fatty acids and used treatments – as for Table 3, same apply in other Tables/figures
  2. According to the Reviewer’s suggestion, manuscript was edited and the captions of tables/figures were improved.

  1. Also avoid using abbreviations in headings and subheadings.
  2. According to the Reviewer’s suggestion, manuscript was edited and headings and subheadings were checked.

Reviewer 2 Report

Authors have modified manuscript as the observations required. From my point of view, the manuscript needs only little (minimal) revision as I reported below.

Material and methods – 2.2.3.3. Fatty acids composition

Authors should specify injection mode. Was it split or split less? If it is split, they should report the split ratio.

Tables

Data that did not showed statistical differences could not be represented with letters (when the letter is the same for all thesis). Authors and editor could evaluate this observation.

Author Response

Reviewer 2

Authors have modified manuscript as the observations required. From my point of view, the manuscript needs only little (minimal) revision as I reported below.

Material and methods – 2.2.3.3. Fatty acids composition

  1. Authors should specify injection mode. Was it split or split less? If it is split, they should report the split ratio.
  2. According to the Reviewer’s suggestion, we have add the split ratio in the injection mode of fatty acids

Tables

  1. Data that did not showed statistical differences could not be represented with letters (when the letter is the same for all thesis). Authors and editor could evaluate this observation.

2. Thanks to the Reviewer’s suggestion, tables were edited.